# Evaluation of Wheat Resistance to Snow Mold Caused by *Microdochium nivale* (Fr) Samuels and I.C. Hallett under Abiotic Stress Influence in the Central Non-Black Earth Region of Russia

**DOI:** 10.3390/plants11050699

**Published:** 2022-03-04

**Authors:** Sulukhan K. Temirbekova, Ivan M. Kulikov, Mukhtar Z. Ashirbekov, Yuliya V. Afanasyeva, Olga O. Beloshapkina, Lev G. Tyryshkin, Evgeniy V. Zuev, Rima N. Kirakosyan, Alexey P. Glinushkin, Elena S. Potapova, Nazih Y. Rebouh

**Affiliations:** 1All-Russian Research Institute of Phytopathology, Bolshye Vyazyomy, Odintsovo District, 143050 Moscow, Russia; sul20@yandex.ru (S.K.T.); glinale1@mail.ru (A.P.G.); 2Federal Horticultural Center for Breeding, Agrotechnology and Nursery, 115598 Moscow, Russia; vstisp@vstisp.org (I.M.K.); yuliya_afanaseva_90@bk.ru (Y.V.A.); 3Department of Agronomy and Forestry, Faculty of Agronomy, Manash Kozybayev North Kazakhstan University, 150000 Petropavlovsk, Russia; mukhtar_agro@mail.ru; 4Department of Biotechnology, Moscow Timiryazev Agricultural Academy, Agrarian University, 127550 Moscow, Russia; beloshapkina58@mail.ru (O.O.B.); r.kirakosyan@rgau-msha.ru (R.N.K.); dac.taer2010@yandex.ru (E.S.P.); 5N.I. Vavilov All-Russian Institute of Plant Genetic Resources (VIR), Bolshaya Morskaya Str. 42–44, 190000 Saint Petersburg, Russia; tyryshkinlev@rambler.ru (L.G.T.); e.zuev@vir.nw.ru (E.V.Z.); 6Department of Environmental Management, Peoples’ Friendship University of Russia (RUDN University), 6 Miklukho-Maklaya Street, 117198 Moscow, Russia

**Keywords:** biotic stresses, fungal pathogens, pink snow mold, VIR gene pool, wheat breeding

## Abstract

*Microdochium nivale* is one of the most harmful fungal diseases, causing colossal yield losses and deteriorating grain quality. Wheat genotypes from the world collection of the N.I. Vavilov Institute (VIR) were evaluated for fifty years to investigate their resistance to biotic stress factors (*M. nivale*). Between 350 to 1085 of winter wheat genotypes were investigated annually. Ten out of fifty years were identified as rot epiphytotics (1978, 1986, 1989, 1990, 1993, 1998, 2001, 2003, 2005 and 2021). The wheat collection was investigated by following the VIR methodological requirements and CMEA unified classification of *Triticum aestivum* L. The field investigations were carried out in the early spring during fixed-route observations and data collection was included on the spread and development degree of the disease, followed by microbiological and microscopic pathogen identifications. The observations revealed that the primary reason for pink snow mold to infect the wheat crops was abiotic stress factors, such as thawed soil covered in snow that increased the soil temperature by 1.0–4.6 °C above normal. Under these conditions, the plants kept growing, quickly exhausting their carbohydrate and protein resources, thus weakening their immune systems, which made them an easy target for different infections, mainly cryophilic fungi, predominantly *Microdochium nivale* in the Moscow region. In some years, the joint effect of abiotic and biotic stresses caused crop failure, warranting the replanting of the spring wheat. The investigated wheat genotypes exhibited variable resistance to pink snow mold. The genotypes Mironovskaya 808 (k-43920) from Ukraine;l Nemchinovskaya 846 (k-56861), from Russia; Novobanatka (k-51761) from Yugoslavia; Liwilla (k-57580) from Poland; Zdar (UH 7050) from the Czech Republic; Maris Plowman (k-57944) from the United Kingdom; Pokal (k-56827) from Austria; Hvede Sarah (k-56289) from Denmark; Moldova 83 (k-59750) from Romania; Compal (k-57585) from Germany; Linna (k-45889) from Finland and Kehra (k-34228) from Estonia determined the sources, stability and tolerance to be used in advanced breeding programs.

## 1. Introduction

Winter crop rot is a complex process that is observed in plants spending a long time at a temperature close to 0 °C in relatively warm soil, without sunlight and under a thick snow cover [1,2]. Under such conditions, the plants quickly burn the nutrients accumulated in their leaves and tillering nodes, hence weakening their nonspecific resistance, and become an easy target for fungal infection, most commonly snow mold and sclerotinia [3].

In mycology, low-temperature fungal pathogen spreading under snow cover had been termed ‘psychrophilic’ until Hoshino, Matsumoto [4] coined another name, ‘cryophilic’, meaning the fungi spend a part or their whole life cycle (in the sexual or asexual state) in a cryosphere, when the biosphere is either constantly or seasonally covered with snow and ice. According to the literature, the pathogenic fungi infecting plants under snow cause a disease known as “snow mold”.

Snow mold, caused by *Microdochium nivale* (Fr) Samuels and I.C. Hallett, *Microdochium*
*majus* (Wollenw.) Glynn and S.G. Edwards, *Typhula idahoensis*, *T. ishikariensis*, *T. incarnata*, *Myriosclerotinia borealis*, *Pythium iwayami* and *P. okanoganense*, is a devastating disease affecting a broad range of small grain cereals [5,6]. *M. nivale* and *M. majus* are the most harmful species among the snow mold-causing pathogens. Pink snow mold caused by *M. nivale* is associated with leaf and stem desiccation, extensive growth of white or pink mycelium and the formation of orange sporodochia, which is expressed in bleached-to-orange-brown patches of matted leaf tissue [7]; this fungal pathogen is frequently noticed in grain cereals, affecting seed germination, as well as influencing the pre- and post-emergence death of seedlings, leading to significant yield losses [8,9,10].

Control of the *M. nivale*, therefore, in most cases is carried out by seed treatment, using fludioxonil and demethylation-inhibiting fungicides, such as difenoconazole, tebuconazole, prothioconazole, bitertanol, etc. [11,12]. Oliver et al. 2013 [13] investigated the effect of microwaves on the fungal pathogens of winter wheat and showed that microwaving significantly reduced *M. nivale* contamination levels on seeds. In addition, the control of *M. nivale* can be achieved by employing crop rotation [14]. In fact, the production of volatile sulfur compounds and soil microbial community composition seems to be effective in controlling *M. nivale*, as demonsrated by Vito et al. [15]. The authors found that the incidence and severity of *M. nivale* were reduced using *Brassica carinata* as precursor crops.

It is well known that the quantity of fertilizers has a significant impact on root rot disease reduction [16,17]. In fact, fertilization, as a conventional agricultural practice, influences the microflora of the soil by changing its chemical composition and physical characteristics. Dordas, 2009 [18] demonstrated that the mode of action of *M. nivale* differed according to nitrogen (N), phosphate (P) and potassium (K) inputs. Previous studies have shown that the high levels of nitrogen (N) applied seem to decrease the severity of root rot infections by enhancing plant physiology [19]. Potassium, when applied at optimal rates, is also known to have the capacity to reduce the host plant’s susceptibility to fungal infections. However, phosphate’s role in plant resistance to pathogens varies considerably [20].

Wheat breeding, so far, has managed to increase grain yield as well as resistance to biotic and abiotic stresses factors. Breeding for disease resistance in wheat has a long and largely successful history [21]. In fact, many studies have demonstrated that the varietal selection and breeding methods are the most reliable and cost-effective method of controlling diseases, especially *M. nivale* [22,23], by developing new cultivars of wheat with disease-resistant characteristics. Plant breeding methods, such as backcross, screening germplasm for resistance sources, the hybridization of selected parents, the selection and evaluation of hybrids and the testing and release of new varieties, are effective methods to transfer one or a few genes controlling a specific trait from one line into a second, usually elite, breeding line, despite the difficulties encountered in terms of time and resources [24,25].

In Russia, the N.I. Vavilov Institute (VIR) is making much effort to develop new varieties focused on varietal resistance to diseases, particularly *M. nivale*, which causes quantitative and qualitative damage to wheat due to the climatic specificity of the central non-black earth region that promotes the development of this disease. The wheat genotypes from the (VIR) world collection can be an important source of resistance to both abiotic and biotic stress factors [3,26]. The purposes of this study is to investigate the gene pool of winter wheat genotypes from the world collection of VIR and identify the sources of resistance to *M. nivale* for future breeding programs.

## 2. Results

The results of the current study represent exploration of the winter wheat gene pool from the VIR world collection in order to detect new sources of resistance to *Microdochium nevale*. During the epiphytotic years, the dominated fungal pathogen causing snow mold in the Moscow region was *M. nivale*. The fungus’s conidial (anamorphic) stage belongs to the Hyphomycetes class, Tuberculariales order, Tuberculariacea family and *Microdochium* genus. Its teleomorph (ascigerous) stage—*Monographella nivalis* var. *nivalis*—belongs to the Ascomycota *phylum*, Euascomycotinae subphylum, Hypocreales order, Nectriaceae family and Monographella genus. The fungus has colorless (pink in mass) fusiform conidia of 14–25 × 3–4 µm in size (Figure 1). Except for conidial sporulation, it also forms an ascigerous stage of surface perithecia found in the bottom part of a plant’s stem, and its ascospores infect the upper leaves in spring and summer when the weather is wet and cool. The fungus’s asci have a thin claviform envelope of 50–70 × 8–10 µm and contain 6–8 spores. Mature ascospores are smooth, of straight or bent ellipsoid shape and have from 1 to 3 septa. Before 1980, it was known as *Fusarium nivale*. Since the beginning of 1980 it has been known as *M. nivale*. Currently, the fungus is known by its teleomorph stage as *Monographella nivalis* var. *nivalis* [4].

The observations over the years allowed to point out 10 years of rot epiphytotics in the central non-black earth region of Russia, namely 1977–1978, 1985–1986, 1988–1989, 1989–1990, 1992–1993, 1997–1998, 2000–2001, 2002–2003, 2004–2005, 2020–2021. The short description of these years is as follows:

**Year 1977–1978**. Snow fell on thawed soil when the air temperature was 1 °C above the average perennial winter temperature (−9.9 °C). The height of snow cover was 30–40 cm. After the snow melted, collection plants started to die due to progressing snow mold. Only 270 out of 900 genotypes survived. Phytopathological assessment of 630 dead genotypes indicated *M. nivale* as a dominating pink snow mold pathogen.

The following varieties demonstrated higher resistance and tolerance to the pathogen (resistance score 7–9; fungal lesion 10–15%): Mironovskaya 808 (k-43920) from Ukraine; Zarya (k-49916) from Russia; Walde (k-51794) from Sweden; Hadmersleben 21687/69 (k-50455), Felakrone (k-50768) from Germany; Antti (k-42673), Jo 832 (k-44775) from Finland; Progress (k-50584), Rizynska 105/71 (k-50603) from Czechoslovakia; Aria (k-50723) from Poland. The genotypes that did not survive mainly came from the countries of Western Europe and Russia’s Krasnodar Region.

**Year 1985–1986**. This winter period was characterized by a high snow cover of 40–50 cm. The snow-cover thawed soil and the season was intermitted by thawing weather (2–4 °C), so the soil had not frozen up till the middle of the winter. In the third week of January, snowing was sometimes intermitted by rain, and the temperature increased by 3 °C from the normal January temperature of −11 °C. Under such conditions, to reduce the temperature in the upper layer of the soil, a snow packing procedure was performed. The long-lasting period of positive temperatures had exhausted the plants and facilitated pink snow mold spread by the end of the winter that was demonstrated by phytopathological analysis. Of 500 genotypes, 353 died, and 132 demonstrated enough tolerance to the disease. These included: Sv 75,268 (k-56156), Helge (k-56872), WW23977 (k-56875), Hildur (k-54130), Sv VG74393 (k-56065), Sv 01,744 (k-56159) from Sweden; TAW37564 (k-55939), TAW5127.72 (k-55940) from Germany; L’vovskaya ostistaya (k-55767) from Ukraine; Novobanatka (k-51761) from Yugoslavia; RMO (k-55220) from Poland.

**Year 1988–1989**. The winter was very warm, so the average monthly temperature in January–March exceeded the norm (8.9 °C) by 9.1 °C. A rare phenomenon of unfrozen soil covered by a thick (60 cm) snow layer was observed. Despite the packing, the plants succumbed to the pathogen; from 500 genotypes only 350 survived. The highest resistance and tolerance to the pathogen was demonstrated by some highly productive genotypes of 450–630 g/m^2^ in crop yield and 40–47 g in 1000-grain weight, such as the Mironovskaya standard (808–365 g/m^2^ and 47 g). These included: PP114–74 (k-57618), Liwilla (k-57580) from Poland; Zdar (UH 7050) (k-57255) from the Czech Republic; Maris Plowman (k-57944) from the United Kingdom; Compal (k-56904), Caristerm (k-57610), Fakta (k-57582) from Germany; Helge (k-56872), Salut (k-58035) from Sweden; Raduga (k-50948), Nemchinovskaya 846 (k-56861), Nemchinovskaya 110 (k-56858), Lutescens 497.83 (k-57657) from Russia (Table 1).

However, the genotypes Bu 22 (k-57603), So 1586 (k-57250), So 8123 (k-56913) from Czech Republic; Weihenstekhaner (и-0109306), Taw 34727/75 (k-55188), Turan (u-109808) from Germany, SMH-71 (и-099339) from Poland; Flambeau (l-56761), Rapier (k-56751), Longbow (k-57611) from United Kingdom; Russia (k-54039), Lutescens 1862 (k-58553), Chernozemka 153 (k-57982) from Russia and Dular (k-56156) from the USA were classified as not resistant to snow mold. Their crop yields were 125–366 g/m^2^ (Table 2).

**Year 1989–1990**. Warm weather with positive (2.4 °C above normal) temperatures in every winter month was typical for this season. A snow cover over thawed soil reached 60–70 cm. Often, winter thawings exhausted the plants, causing their deaths. In spring, after the snow melted, the dead plants were found covered in the mycelium and conidial sporulation of pink snow mold. Out of 500, 164 genotypes died, mostly those from the countries of Western Europe and Russia’s Krasnodar region. The highest resistance and tolerance to both abiotic and biotic factors was demonstrated by: Pokal (k-56827) from Austria; Hvede Sarah (k-56289) from Denmark; TAW4279180 (k-58363), Fakta (k-57582), Compal (k-57585), Fakon (k-58187), Kronjuwel (k-57615), TAW 39496.75 (k-56903) from Germany; Venture (k-57231) Longbow (k-57611), Granta (k-57219) from the United Kingdom; Sv75355 (k-56158) from Sweden; Titan (k-58059) from the United States; Erythrospermum 9736 (k-57479), Grekum 9271 (k-57472) from Ukraine; Yantarnaya 50 (k-54610), Nemchinovskaya 52 (k-59269), Belosnezhnaya (k-57573) from Russia. Their crop yields were 449–600 g/m^2^ against 495 g/m^2^ of the Mironovskaya standard (Table 3).

However, the genotypes Talon (k-62743), Peacock (k-62744), Fortress (k-62745), Admiral (k-62747) from United Kingdom, Accent (k-62843), Avir (k-62844), TMillev (k-62845) from Netherlands, Иcтpa-1 62734 from Russia and Hjan Ilves (k-63015), Ilves (k-63016) from Finland were classified as not resistant to snow mold. Their crop yields were 45–245 g/m^2^ (Table 4).

**Year 1992–1993**. In this season, snow covered warm soil and caused an ice layer to form. The temperature during winter months exceeded the average perennial winter norm (−9.9 °C) by 1–2 °C. Seventy percent of the collection plants died after the snow melted and only 40% of the Mironovskaya 808 standard survived. The leaves and stems of the dead plants were covered in pink snow mold. At the level of the standard, a proper resistance to the infection was demonstrated by three genotypes: Moldova 83 (k-59750) from Romania; Sparta (k-60094) from Czechoslovakia; and Borden (k-50317), from Canada.

**Year 1997–1998**. The season was warm, but January and February did not give enough snow and the temperature was 3 °C above normal. The precipitation level was two times below normal. After the snow melted, 50–70% of the collection plants either died or were severely infected by pink snow mold. The highest resistance and tolerance to both abiotic and biotic factors were demonstrated by: Mironovskaya 808 (k-43920) from Ukraine, Compal (k-57585), Tukan (k-57579) from Germany, Zarya (k-49916) from Russia; PE 649 (k-52656) from Finland; Sheriff Dickopf (k-40526), Sv 65646 (k-55305), WW 24089 (k-51803), Sv 61246 (l-47099) from Sweden, Salwa (u-538083) from Poland and Maris Marksman (k-55233) from United Kingdom.

The 1000-grain weight of these varieties varied between 39.3 and 41.5 g, and the crop yields between 290 and 350 g/m^2^, while, in the standards, these parameters were 45–56 g and 330–470 g/m^2^, respectively (Table 5).

However, the genotypes Solid (k-51796) from Sweden, Kalininskaya 11 (k-46957), Erythrospermum 60 (k-50968) from Russia, Hohenthurmer 6831/68 (k-50470), Cyrano (k-51609), 618/67 (k-53517) and Duellant (k-53518) from Germany, WZ 945-13 (k-51713) from Netherlands, TJB 364/636 (k-53486) from United Kingdom and Hybrid (k-53500) from Denmark did not expressed resistance to snow mold. Their crop yields were 80–200 g/m^2^ (Table 6).

**Year 2000–2001**. Thawings and rains were typical for this season in which the average temperature was 2–4 °C above normal. The precipitation level (193.9) also exceeded the norm (111 mm). The soil did not freeze up properly, causing a massive rot in the collection plants, so 60–80% of them died, with pink snow mold prevailing in the genotypes from Western Europe, the CIS countries and Russia. Out of 1058 varieties, the proper level of resistance (7–9 points) and tolerance to both abiotic and biotic factors at a crop yield of 300 –420 g/m^2^ was demonstrated by: Pamyati Fedina (k-62440), Lyutestsens 306 (k-58827), Lyutestsens 319 (k-59267), Ivanovskaya 16 (k-58526) from Russia; Kehra (k-34228) from Estonia; Varmlands (k-34230) from Sweden.

**Year 2002–2003**. In this season, December had a small amount of snow, and January and February were very warm, with the temperature exceeding the norm by 1.9–4.6 °C, which had a negative effect on the winter wheat and caused its premature death. Out of 1200 collection varieties analyzed, the level of resistance and tolerance to pink snow mold at appropriate crop yield was demonstrated by: Pamyati Fedina (k-62440), Raduga (k-50948), Nemchinovskaya 24 (k-65757), Moskovskaya 56 (k-65760), Lutescens 319 (k-59267) and Ivanovskaya 16 (k-58526) from Russia; Varmlands (k-34230) from Sweden; Orestis (k-64034), Bussard (k-64027), Gelderseries (b/k) from Germany; Zenta (k-56825) from Switzerland; Compair (k-51913), Maris Ranger (k-50721), Rothwell Senator (k-50762) from the United Kingdom; standard Mironovskaya 808 (k-43920) from Ukraine-280 g/m^2^ (Table 7).

However, the genotypes KOS 1853/92 (k-63407), KOS 2696-93 (k-63420) from Russia, Chacay (k-63426) from Chile and Betta (k-63430) from South Africa were classified as not resistant to snow mold, with crop yields of 45–120 g/m^2^ (Table 8).

**Year 2004–2005**. In this season, snow covered thawed soil and the temperature exceeded the norm by 2 °C. The height of the snow cover was 20–25 cm, causing ice layer formation. In spring, the plants died en mass due to pink snow mold infection. The phytopathological analysis detected *M. nivale* as the reason for death in 528 wheat varieties. The highest degree of resistance and tolerance were demonstrated by: Kazanskaya 560 (k-63565), Karel’skaya bezostaya (k-40579) from Russia; k-15339 (Belarus), Tab 2598 (k-44326) from Finland. The standard was the variety Moskovskaya 39, with a 1000-grain weight of 39.7 g (Table 9).

However, the genotypes Tomo (k-63912) from United Kingdom, Rentol (k-64012) from Sweden and Czech Republic (k-63892) from South Africa did not express resistance to snow mold, with crop yields of 80–160 g/m^2^ (Table 10).

**Year 2020–2021**. This year’s snowfall was preceded by icy rain that fell on thawed soil. The temperature of the winter months was 1–3 °C above normal. The height of the snow cover reached 90–100 cm, causing ice layer formation. After the snow melted in spring, a significant spread of pink snow mold was observed, e.g., in the precision agriculture experiment carried out at Russian State Agrarian University-Moscow Timiryazev Agricultural Academy; the spread reached 94% in the Timiryazevska-–Yubileinaya winter wheat and was not a focal phenomenon but covered almost the whole field [27].

In the year 2020–2021, the pink snow mold infection rate exceeded 10 times the average for the 50-year period due to the ice layer that formed following the snow that covered the thawed soil. The typical symptoms of the disease include stem bending, yellowing, leaf gluing and perishing (starting from peripheral ones) and plant death. The preserved leaves often had watery spots with a web-like mold of mycelium and conidial sporulation. After the leaves perished (Figure 2), the fungus attacked the tillering node (Figure 2a). In some years, microscopic investigation of dead plants made it possible to detect teleomorph sporulation (reddish rounded perithecia). In the extreme epiphytotic years (1993, 2001), the crop fields often had empty spots covered in perished leaves (Figure 2b) that sometimes occupied significant areas [28].

## 3. Discussion

According to Bruehl. 1982 [29], the protracted snow cover forms a dark, humid environment with constant temperatures which reduces plant metabolism and prevents photosynthesis, thus decreasing the plants’ immunity as well as promoting the development of psychrophilic fungi called snow molds, especially *Microdochium nivale*.

In the present study, annually, during the winter months, samples of soil with plants from the field were taken to analyze the condition of plants under snow cover in the laboratory. Over all the epiphytotic years, the depletion of plants due to their prolonged stay under the snow cover was noted, and pink snow mold damage was recorded only during early spring examination after snow melting. However, the dynamics of the development and the intensity of the plant infestation by the pathogen were not revealed. Moreover, the soil samples taken from the end of March to the beginning of April have revealed the presence of spores on the plants weakened by the abiotic stress. For that reason, we consider that the snow mold is a complex pathological process, initially launched by abiotic stress, followed by cryophilic fungal infection starting with sub-saprotrophic feeding on the weakened plants. Furthermore, many authors have studied the relationship between frost tolerance and snow mold resistance and have demonstrated that the resistance to low temperatures induces snow mold resistance [2,30]. In fact, for cryophilic fungi, the plants with disrupted physiological processes due to low temperatures, such as excessive breathing and premature decomposition of hydrocarbons into sugars and proteins into amides and amino acids, are an ideal nutritious substrate [3,31].

Annick et al. [23] showed that the tolerance of plants to frost allows them to acquire resistance to snow mold due to the high levels of carbohydrates after cold acclimation in frost-tolerant pants compared with sensitive plants. Pociecha et al. [32] demonstrated that the changes in biochemical parameters of the studied plants were observed after cold acclimation. Furthermore, Hiilovaara-Teijo et al. [33] found that, in response to cold, the resistant winter rye accumulated more proteins in the leaf apoplast. These results are in accordance with those reported in the present study and confirmed that snow mold infection start by sub-saprotrophic feeding on the weakened plants by biotic stress. Accordingly, we report that the joint effect of abiotic and biotic stresses sometimes causes complete crop failure, as it happened in 1993 and 2001.

The results of this 50-year study also allowed us to conclude that, for 40 years, the winter wheat overwintering was mainly affected by frost (abiotic stress). In this case, a snow layer covered not thawed but frozen soil, which is typical central non-black earth region of Russia. These years were cold, and their temperature varied within −12–35 °C. Under such conditions, no pink snow mold infection was observed. It is also noteworthy that for these years, the studied soil samples from winter and early spring times did not contain the pathogen and plants were not chlorotic, hence, they were not physiologically exhausted. In plate experiments, the optimum temperature for growth of *M. nivale* are around −6.3 °C and −2.2 °C respectively [34].

The development of pink snow mold in the 2020–2021 growing year indirectly indicates the arrival of global climate warming on the planet, and it is possible that winter conditions similar to those prevailing in 2020–2021 will be frequent in the future. For that reason, plant breeders have to be ready to account for sudden warmings. Our study demonstrated that during the epiphytotic years, snow covering thawed soil made the winter wheat plants burn their nutrient reserves and continue the vegetation process to complete exhaustion. As a result, the plants weakened their immune systems and were unable to restore since photosynthesis is not possible without sunlight, i.e., they died of exhaustion due to the lack of sunlight, having depleted their carbohydrate and protein reserves. At that moment, cryophilic fungi became harmful, dominated by *M. nivale*, a pink snow mold pathogen causing mass crop mortality. When emerging from under the snow, some samples were found in the lesions with dead grayish and brownish-yellow leaves and with a pink coating on the leaves and the seed (Figure 2a). It is still discussable whether the fungicide spraying recommended by several researchers and performed in spring after snow melts can be effective [35]. In our experience, at a high infection rate, it does no good and further suppresses the exhausted plants (Figure 2b). The only way to reduce the damage and save the plants is through agrotechnical methods combined with nitrogen fertilization in spring.

Soil and plant residuals are the main source of infection. The pathogen can spread along a soil surface and can get into soil, especially when the temperature is low. In the soil, the fungus’s conidia survive for 30 days or more, as demonstrated by Couteaudier et al. [36]. However, it is noteworthy that they become active only when the plants are weakened by abiotic stresses. In this respect, *Microdochium nivale* can be defined as one of the most harmful pathogens causing pink snow mold in the winter wheat under the central non-black earth region of Russia conditions. In view of the lack of relevant research to distinguish cultivars resistant to *M. nivale*, this study has been directed to the investigation of the gene pool of winter wheat from the VIR world collection to estimate their resistances and tolerances to this complex conjugate condition. It has enabled us to detect highly resistant and tolerant cultivars originating from different countries and having the resistance rate of 7–9 points at the prevalence rate of 10–15%. These include: Linna (k-45889), Hja 24499 (k-62273) from Finland; WW 23262 (k-51808), Holger (k-62310), Sheriff Dickopf (k-40526), Svalofs Sonnet II (k-45132), Sv 6246 (k-47099), WW 24089 (k-51803), Konsul (k-64011), Sv 65646 (k-55305), Hildur (k-54130), Kosack (k-58137) from Sweden; Bijon (k-59520), Otto (k-59527) from Belgium; PP 114-74 (k-57618), Boxer (k-59536), Beauford (k-63920), Liwilla (k-57580) from Poland; Hornet (k-60100), Legend (k-61498), Maris Plowman (k-57944), Wizard (k-57229) from the United Kingdom; PE 6490 (k-52656), Trifolium 33 (k-56290) from Denmark; Zdar (k-57255) from the Czech Republic; Remus (k-56904), Caristerm (k-57610), Tukan (k-57579), Zentos (k-64030), Fazit (k-64032), Bussard (k-64027), Faktor (k-64028), Olimp (k-61455), Aron (k-64007), Apollo (k-61463), Criewener 2865/69 (k-52862) from Germany; Pammets (k-26236) from Norway; Raduga (k-50948), Nemchinovskaya 846 (k-56861), Nemchinovskaya 52 (k-59269), Lutescens 497/83 (k-57657), Shatilovskaya (k-37478), Lutescens 103 (k-55955), Lutescens 99 (k-55956), Sibirskaya Ul’yanovka (k-56057), MRt-833 (k-59228), MRt-340 (k-59233), M 15/6 ostistyy (k-59239), MGs-2287-t (k -60071) from Russia; Suzorie (k-59245) from Belarus; Brigantine (k-55181) from Ukraine.

The winter wheat variety Moskovskaya 39 was created on the basis of the genotypes selected during this study. Polityko et al. [37] investigated the potential of winter wheat variety Moskovskaya 39. The authors showed that higher values, in term of grain quality, weight of 1000 grains, gluten and protein content in grain, flour strength, porosity and resistance to fungal diseases, especially to *M. nivale*, were observed in the variety Moskovskaya 39 among seven studied varieties of winter wheat.

In the 21st century, the climate is expected to warm in Russia, the countries of Western Europe and Africa [38], so we recommend the potential sources of high resistance and tolerance to *Microdochium nivale* to be used in these countries for breeding adaptive varieties that produce a large yield of high-quality grain.

## 4. Material and Methods

### 4.1. Experimental Site

The field experiment was carried out at the Federal Horticultural Center for Breeding, Agrotechnology and Nursery, in the department of the gene pool and biological resources of plants, the former Moscow branch of the VIR. It is located in the village of Mikhnevo, Moscow region, in the central part of the southern Taiga forest soil-climatic zone (central non-black earth region of Russia).

### 4.2. Climatic Conditions and Soil Characteristics

The climate of Moscow region is moderately continental and humid. The average annual precipitation was 450–800 mm, and the moisture was sufficient in years with normal precipitation. The probability of excessively wet years was 25–40%, and that of arid and semiarid years about 12–20%. Accumulated temperatures above 10 °C decreased from 2100° in the east and south-east to 1900° in the northwest, and vegetation periods (above 10 °C) decreased accordingly from 140–145 to 120–125 days [39].

About 70% of precipitation was recorded in the warm season, which ensures favorable conditions for plant growth and development. Soils freeze to the depths of 50–75 cm in open areas and 30–50 cm in sheltered areas. Full thawing of soil is usually expected from April 21 to 29. Soil tilth was achieved on May 20 in loam soils, and on May 18 in sandy loam soils. The frost-free period usually lasts 120–135 days, which allows cultivated plants to achieve full ripeness. Permanent snow cover, with an average depth of 35 cm from 25 November to 2 December, persisted up to 137–143 days. The hydrothermal index was 1.3–1.4. North- and southwest were the prevailing wind direction trends in Moscow region throughout the year.

### 4.3. Methodical Approach Applied in Plant Breeding

Winter wheat samples were introduced into a well-defined crop rotation in late August with black fallow crops. An SSFK-7M seeder was used, and the sowing density was 500 grains per m^2^. Mineral fertilizers were applied during the land preparation period at the following rates: 68 kg/ha for nitrogen, 60 kg/ha for phosphorus and of 30 kg/ha for potassium. Furthermore, 50 kg/ha of nitrogen was applied as top dressing in spring. The reference cultivars Mironovskaya 808 (k-43920, Ukraine) and, in some years, Polukarlik 3 (k-54508, Ukraine), and te Moscow cultivars, such as Zarya (k-49916), Nemchinovskaya 52 (k-59269), Moskovskaya 39 (k-64160), were planted with intervals of 10 and 50 samples, respectively. The wheat collection was studied in compliance with the VIR Methodical guidelines [40] and the International COMECON list of descriptions for genus *Triticum* L. [41].

Every year, 350–1085 genotypes of winter wheat were studied, the precursor was complete fallow, the data collected and analyzed annually were germination, overwintering, snow mold damage, frost resistance, *Microdochium nivale* infestation, vegetation period, plant height, lodging and the weight of 1000 grains. Yield and grain quality were measured only for the distinguished varieties. Every year, genotypes distinguished by resistance to *Microdochium nivale* were statistically evaluated with a standard variety based on yield, weight of 1000 grains and plant height.

The winter agroclimatic conditions of snow mold epiphytotic years (1978, 1986, 1989, 1990, 1993, 1998, 2001, 2003, 2005, 2021) in comparison with perennial indications are given in Figure 3 and Figure 4.

### 4.4. Diseases Infestation Scale

Snow mold was accounted for in two periods—in autumn, when the tillering phase was reached, and in spring, after snow melting. At least 10 points were selected for accounting at the experimental sites. Visual examination of the sowing was carried out at regular intervals, the number of affected plants per 1 m^2^ and the degree of their infestation were noted, as well as the form of the disease: diffuse or focal. At each point, 50 plants were selected and evaluated in the laboratory according to the appropriate scale: 0—absence of disease, 1—less than 10% of leaves were affected, 2—the lower leaves were completely affected, or more than 30% of the plant was affected, 3—the lower and upper leaves were affected, the degree of damage was more than 70%, shoots were dying, 4—all leaves and shoots were affected, the plant died [41,42].

## 5. Conclusions

Our perennial investigation and analysis of the results of pink snow mold spread in winter wheat during a number of epiphytotic years has demonstrated that the primary reason for the spread happens to be such abiotic stress factors as the snow covering thawed soil at ambient temperature exceeding the norm (−9.5 °C) by 1.0–4.6 °C. Under such conditions winter wheat sprouts keep growing, breathing and burning their nutrient reserves (proteins and carbohydrates). Meanwhile, their immune system gets weaker, rendering the plants susceptible to infections, which are mainly caused by the cryophilic fungi such as *Microdochium nivale.* The joint effect of abiotic and biotic stress factors results in excessive of crop mortality that it sometimes requires spring wheat reseeding. As a protective measure, we recommend combining agrotechnical methods with nitrogen fertilization in spring. The potential sources of resistance and tolerance to rot and pink snow mold resulting from this study can be recommended to plant breeders of different countries for selection.

## Figures and Tables

**Figure 1 plants-11-00699-f001:**
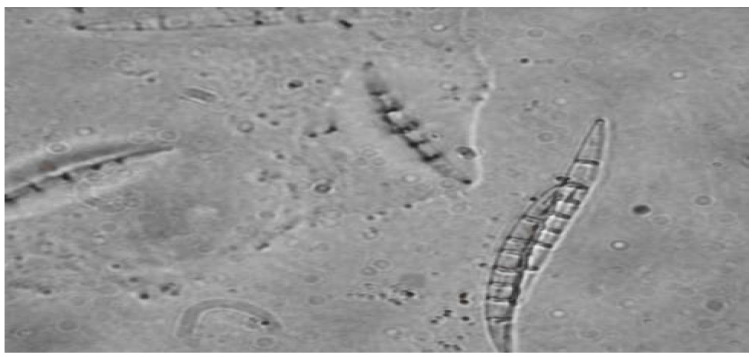
*Microdochium nivale* (Fr.) Samuels and I.C. Hallett (a.k.a *Fusarium nivale* (Fr.) Ces.ExBerl. and Voglino) conidia (scale bar: 10 µm).

**Figure 2 plants-11-00699-f002:**
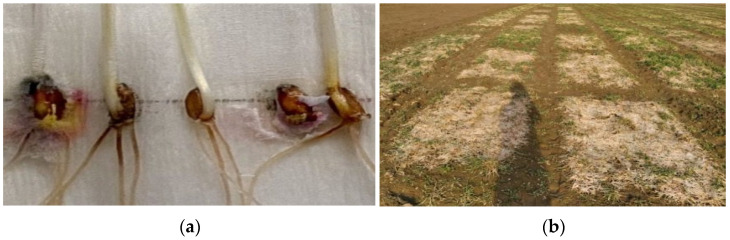
(**a**)—Grains infected by *Microdochium nivale*; (**b**)—an experimental field infected by *Microdochium nivale*, 2021.

**Figure 3 plants-11-00699-f003:**
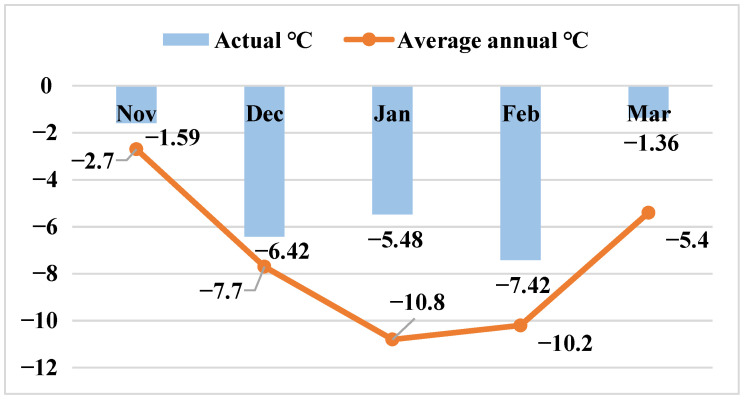
Average monthly temperatures in the epiphytotic years.

**Figure 4 plants-11-00699-f004:**
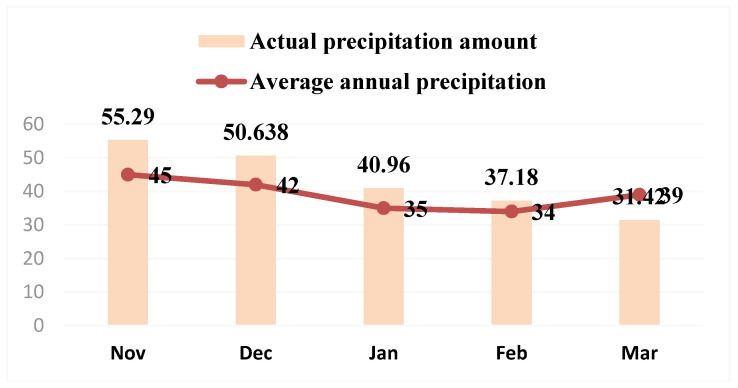
Average monthly precipitation rates in the epiphytotic years.

**Table 1 plants-11-00699-t001:** Yield of the resistant genotypes (1988–1989).

Resistant Genotypes		
Genotypes	Country	Yield g/m^2^
Mironovskaya st	Ukraine	365
PP114-74 (k-57618)	Poland	615
Liwilla (k-57580)	Poland	630
Zdar (UH 7050)	Czech Republic	595
Maris Plowman (k-57944)	United Kingdom	545
Caristerm (k-57610)	Germany	585
Fakta (k-57582)	Germany	575
Compal (k-57585),	Germany	565
Helge (k-56872)	Sweden	450
Salut (k-58035)	Sweden	550
Raduga (k-50948)	Russia	570
Nemchinovskaya 846 (k-56861)	Russia	630
Nemchinovskaya 110 (k-56858)	Russia	610
Lutescens497.83 (k-57657)	Russia	590

**Table 2 plants-11-00699-t002:** Yield of the non -resistant genotypes (1988–1989).

Non-Resistant Genotypes		
Genotypes	Country	Yield g/m^2^
Bu 22 (k-57603)	Czech Republic	245
So 1586 (k-57250)	Czech Republic	315
So 8123 (k-56913)	Czech Republic	290
Weihenstekhaner (и-0109306)	Germany	125
Taw 34727/75 (k-55188)	Germany	335
Turan (u-109808)	Germany	315
SMH-71 (и-099339)	Poland	325
Flambeau (l-56761)	United Kingdom	315
Rapier (k-56751)	United Kingdom	340
Longbow (k-57611)	United Kingdom	330
Russia (k-54039)	Russia	308
Lutescens 1862 (k-58553)	Russia	366
Chernozemka 153 (k-57982)	Russia	305
Dular (k-56156)	USA	320

**Table 3 plants-11-00699-t003:** Yield of the resistant genotypes (1989–1990).

Resistant Genotypes		
Genotypes	Country	Yield g/m^2^
Mironovskaya st	Ukraine	495
Pokal (k-56827)	Austria	520
Hvede Sarah	Denmark	537
TAW4279180 (k-58363)	Poland	496
Fakta (k-57582)	Germany	590
Compal (k-57585),	Germany	595
Fakon (k-58187)	Germany	511
Kronjuwel (k-57615)	Germany	600
TAW 39496.75 (k-56903)	Germany	548
Venture (k-57231)	United Kingdom	544
Longbow (k-57611)	United Kingdom	509
Granta (k-57219)	United Kingdom	490
Sv75355 (k-56158)	Sweden	527
Titan (k-58059)	USA	501
Erythrospermum 9736 (k-57479)	Ukraine	449
Grekum 9271 (k-57472)	Ukraine	513
Yantarnaya 50 (k-54610)	Russia	482
Nemchinovskaya 52 (k-59269)	Russia	565
Belosnezhnaya (k-57573)	Russia	468

**Table 4 plants-11-00699-t004:** Yield of the non-resistant genotypes (1989–1990).

Non-Resistant Genotypes		
Genotypes	Country	Yield g/m^2^
Talon (k-62743)	United Kingdom	245
Peacock (k-62744)	United Kingdom	45
Fortress (k-62745)	United Kingdom	70
Admiral (k-62747)	United Kingdom	145
Accent (k-62843)	Netherlands	135
Avir (k-62844)	Netherlands	95
Millev (k-62845)	Netherlands	45
Иcтpa-1 62734	Russia	50
Hjan Ilves (k-63015)	Finland	115
Ilves (k-63016)	Finland	125

**Table 5 plants-11-00699-t005:** Yield of the resistant genotypes (1997–1998).

Resistant Genotypes		
Genotypes	Country	Yield g/m^2^
Mironovskaya st	Ukraine	350
Compal (k-57585)	Germany	350
Tukan (k-57579)	Germany	310
Zarya(k-49916)	Russia	290
Sheriff Dickopf (k-40526)	Sweden	370
Sv 65646 (k-55305)	Sweden	346
WW 24089 (k-51803)	Sweden	345
Sv 61246 (l-47099)	Sweden	342
Jo 01177 (k-44704)	Finland	320
PE 649 (k-52656)	Denmark	335
Salwa (u-538083)	Poland	350
Maris Marksman (k-55233)	United Kingdom	370

**Table 6 plants-11-00699-t006:** Yield of the non-resistant genotypes (1977–1998).

Non-Resistant Genotypes		
Genotypes	Country	Yield g/m^2^
Solid (k-51796)	Sweden	200
Kalininskaya 11 (k-46957)	Russia	90
Erythrospermum 60 (k-50968)	Russia	175
Hohenthurmer 6831/68 (k-50470)	Germany	165
Cyrano (k-51609)	Germany	150
618/67 (k-53517)	Germany	130
Duellant (k-53518)	Germany	125
WZ 945-13 (k-51713)	Netherlands	145
TJB 364/636 (k-53486)	United Kingdom	80
Hybrid (k-53500)	Denmark	95

**Table 7 plants-11-00699-t007:** Yield of the resistant genotypes (2002–2003).

Resistant Genotypes		
Genotypes	Country	Yield g/m^2^
Mironovskaya st	Ukraine	280
Pamyati Fedina (k-62440)	Russia	420
Raduga (k-50948)	Russia	910
Nemchinovskaya 24 (k-65757)	Russia	450
Moskovskaya 56 (k-65760)	Russia	400
Lutescens 319 (k-59267)	Russia	422
Ivanovskaya 16 (k-58526)	Russia	367
Varmlands (k-34230)	Sweden	302
Orestis (k-64034)	Germany	310
Bussard (k-64027)	Germany	400
Gelderseries (b/k)	Germany	490
Zenta (k-56825)	Switzerland	280
Compair (k-51913)	United Kingdom	965
Maris Ranger (k-50721)	United Kingdom	930
Rothwell Senator (k-50762)	United Kingdom	900

**Table 8 plants-11-00699-t008:** Yield of the nonresistant genotypes (2002–2003).

Non-Resistant Genotypes		
Genotypes	Country	Yield g/m^2^
KOS 1853/92 (k-63407)	Russia	120
KOS 2696-93 (k-63420)	Russia	45
Chacay (k-63426)	Chile	55
Betta (k-63430)	South Africa	60

**Table 9 plants-11-00699-t009:** Yield of the resistant genotypes (2004–2005).

Resistant Genotypes		
Genotypes	Country	Yield g/m^2^
Moskovskaya 39 st	Russia	250
Kazanskaya 560 (k-63565)	Russia	327
Karel’skaya bezostaya (k-40579)	Russia	160
k-15339	Belarus	295
Tab 2598 (k-44326)	Finland	187

**Table 10 plants-11-00699-t010:** Yield of the nonresistant genotypes (2004–2005).

SNon-Resistant Genotypes		
Genotypes	Country	Yield g/m^2^
Tomo (k-63912)	United Kingdom	80
Rentol (k-64012)	Sweden	110
Siria (k-63892)	Czech Republic	160

## Data Availability

Not applicable.

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
