# Peer review of "Evaluation of Wheat Resistance to Snow Mold Caused by Microdochium nivale (Fr) Samuels and I.C. Hallett under Abiotic Stress Influence in the Central Non-Black Earth Region of Russia"

_plants, 2022, doi:10.3390/plants11050699_

Round 1
Reviewer 1 Report
This manuscript brings data on the resistance of winter wheat genotypes to snow mold caused by Microdochium nivale in Russia. Although the study is based on a large number of plants studied, statistical analysis of the data is lacking. At least some graphical representation of the results would be useful, e.g. the number or percentage of susceptible and resistant wheat genotypes in epiphytotic years; or the number/proportion of wheat genotypes showing different resistance levels.
The manuscript needs some improvement, but it is based on solid data and can be valuable to readers.
The Methodology section should be more detailed, no information on microbiological and microscopic pathogen identification is provided. It is not clear whether artificial inoculations of plants were made or natural infections were evaluated. If artificial infections have been made, add the inoculation method; in the case of a natural infection, indicate how you distinguished infections caused by a morphologically similar species Microdochium majus.
In the Results, there is information about the experiment carried out at the Russian State Agrarian University - Moscow Timiryazev Agricultural Academy in year 2020-2021, but this experimental site is not mentioned in the Material and methods.
The Discussion must be rewritten, the results are not discussed adequately. The paragraphs describing the anamorphic and teleomorphic stages of the fungus (Lines 263-276) should not be in the Discussion. Figure 1 should be moved in the results. The scale bar in Figure 1a is missing. Figure 2 is redundant because it is similar to Figure 1c.
Correct the authors' names of scientific names of fungi in the manuscript title and throughout the text (authorities can be found in the Index Fungorum database: www.indexfungorum.org). Authorities for all fungal pathogens should be included at first mention of the taxa.
I think the correct English name for the area is "Central Non-Black Earth Region of Russia", use it uniformly in the title and throughout the text.
Check the numbering of all cited papers throughout the text and in the References, because there are mistakes, e.g.
- the publication "Hoshino 2012" has two reference numbers [4] and [28];
- “Bruehl 1982” has reference number [28] in the text (line 224), but [27] in the References;
- "the International COMECON List of Descriptors for the Genus Triticum L.” has the reference number [38] in the text (line 371), but [37] in the References. Correct author name of this publication in the References.
Author Response
Please kindly find attached the corrections.

Reviewer 2 Report
Dear Authors,
congratulations for your longitudinal study. Unfortunately, you did not rely on statistical analysis of your data, therefore the manuscript cannot be presented in this descriptive form, in a scientific journal. You need to undergo a plan of how to analyze your data so that your observations can be backed by the power of mathematics and also to bring to light aspects you omitted due to inner limits of description.
below you may find some comments:
Introduction
Line 52-62: names -> italic
Results:
Line 108: “in order to distinguish” -> to detect
The description of abiotic factors, degree of attack and resistant/tolerant genotypes, and correspondent yields should be inserted in a table. It is really difficult for the reader to follow the information and to get the relevant information.
Line 254: you did not demonstrate you statement unless you have analysis to rely on.
Therefore, you should plan statistical analysis. One option is to make a PCA to analyze the correspondence between variables like “soil covered with snow-small amount/medium/high”, precipitation, temperature, level of attack or “germination, overwintering, snow mold damage, frost resistance, Microdochium nivale infestation, vegetation period, plant height, lodging, and the weight of 1000 grains”. Otherwise, without a proper analysis the study is merely a descriptive one, where you cannot get the power of the statistical analysis. Also, pairwise correlations are a possible way to bring to light your results. Also, data should have a proper statistical analysis to detect the significance of the differences! Try ANOVA or linear mixed models..
Methods
Line 374: please explain the scale used for degree of attack
Author Response
Please kindly find attached the corrections

Round 2
Reviewer 1 Report
Dear Authors,
Thank you for improving your manuscript and for your detailed replies to comments on the submitted version. I have no further comments to add.